# How to Enhance Causal Discrimination of Utterances: A Case on Affective Reasoning

**Hang Chen** and **Xinyu Yang** and **Jing Luo**
Xi'an Jiaotong University
{albert2123,luojingl}@stu.xjtu.edu.cn
yxyphd@mail.xjtu.edu.cn

**Wenjing Zhu**
Du Xiao Man Inc.
zhuwenjing02@duxiaoman.com

## Abstract

Our investigation into the Affective Reasoning in Conversation (ARC) task highlights the challenge of causal discrimination. Almost all existing models, including large language models (LLMs), excel at capturing semantic correlations within utterance embeddings but fall short in determining the specific causal relationships. To overcome this limitation, we propose the incorporation of *i.i.d.* noise terms into the conversation process, thereby constructing a structural causal model (SCM). It explores how distinct causal relationships of fitted embeddings can be discerned through independent conditions. To facilitate the implementation of deep learning, we introduce the *cogn* frameworks to handle unstructured conversation data, and employ an autoencoder architecture to regard the unobservable noise as learnable "implicit causes." Moreover, we curate a synthetic dataset that includes *i.i.d.* noise. Through comprehensive experiments, we validate the effectiveness and interpretability of our approach. Our code is available in https://github.com/Zodiark-ch/mater-of-our-EMNLP2023-paper.

## 1 Introduction

Nowadays, numerous conversation recognition tasks (such as Emotion Recognition in Conversation (ERC) task (Pereira et al., 2023; Thakur et al., 2023), Intent Recognition (IR) task (Ye et al., 2023; Ni, 2023) and Dialogue Act Recognition (DAR) task (Arora et al., 2023)) have shown promising performance in specialized supervised and unsupervised methods. Considering the RoBERTa pretrained model (Liu et al., 2019) as the examples, "My eyelids are fighting" and "I want to sleep," which have similar semantics but different tokens can be well fitted within embeddings. (i.e., these two embeddings exhibit a strong resemblance via certain metrics such as cosine similarity.)

However, when it comes to the relationship between two utterances, denoted as $A$ and $B$, wherein

their embeddings can be fitted, various possible relationships exist: $A$ acts as the cause of $B$ ($A \rightarrow B$), $A$ acts as the outcome of $B$ ($A \leftarrow B$), or more complex, $A$ and $B$ are both influenced by a common cause ($A \leftarrow C \rightarrow B$), and so on. Particularly in reasoning tasks (Uymaz and Metin, 2022; Feng et al., 2022), it is crucial for these methods to transcend the mere fitting of embeddings and possess the capacity to discriminate diverse causal relationships. (i.e., the ability of **causal discrimination**) (Bao et al., 2022; Shirai et al., 2023).

To specifically investigate the causal discrimination capability of existing methods in conversation, we narrow down our research to a particular task: Affective Reasoning in Conversation (ARC), which has included Emotion-Cause Pair Extraction (ECPE) Xia and Ding (2019) and Emotion-Cause Span Recognition (ECSR) Poria et al. (2021).

We begin with conducting tests to evaluate the causal discrimination of existing methods including the large language models (LLMs) (Kasneci et al., 2023). One typical evaluation involves the causal reversal test: for emotion-cause utterance pairs with true labels $(A, B)$ representing a causal relationship of $B \rightarrow A$, we scrutinize the predictions generated by the existing methods using both positive pairs $(A, B)$ and negative pairs $(B, A)$. The results reveal that all the examined methods performed similarly across the two sample types. As we are concerned, they lacked causal discriminability. (Details are shown in Section 2.3)

In order to discriminate different causal relationships between two similar embeddings, we construct the dialogue process as a Structural Causal Model (SCM). Many endeavors (Cheng et al., 2022; Nogueira et al., 2022) supporting that *i.i.d.* noise of SCM could facilitate the discrimination of causal relationships when fitting two variables. Under the presence of noise, each utterance is not only explicitly influenced by the other utterances but also implicitly influenced by the *i.i.d.* exoge-

nous noise. Consequently, this framework ensures that two fitted embeddings result in diverse causal relationships, which are determined by corresponding independent conditions between the residual terms and embeddings. For simplicity, we refer to other utterances as explicit causes and exogenous noise as implicit causes.

Furthermore, to enable the learnability of such causal discrimination within embeddings, we propose a common skeleton, named *centering one graph node (cogn)* skeleton for each utterance derived from some broadly accepted prior hypotheses. It can address the challenges arising from variable-length and unstructured dialogue samples. Subsequently, we develop an autoencoder architecture to learn the unobservable implicit causes. Specifically, we consider the implicit causes as latent variables and utilize a graph attention network (GAT) (Veličković et al., 2017) to encode its representation. Additionally, the decoder leverages the inverse matrix of the causal strength, ensuring an accurate retrieval of the causal relationships.

Finally, we conduct extensive experimental evaluations: 1) our approach significantly outperforms existing methods including prominent LLMs (GPT-3.5 and GPT-4) in two affective reasoning tasks (ECPE and ECSR) and one emotion recognition task (ERC), demonstrating its effectiveness in affective reasoning. 2) our method exhibits a significant reduction in false predictions for negative samples across three causal discrimination scenarios. 3) we curate a synthetic dataset with implicit causes to visualize the latent variable in our implementation.

Our contribution is four-fold:

- We formulated the dialogue process as an SCM and analyzed the causal relationships represented by different independent conditions.

- We devised the *cogn* skeleton to address the problems of variable-length and unstructured dialogue samples.

- We adopted an autoencoder architecture to overcome the unobservability of implicit causes and make it learnable.

- We constructed a synthetic dataset with implicit causes and conducted extensive evaluations of our proposed method.

## 2 Related Works and Challenges

### 2.1 Task Definition

For notational consistency, we use the following terminology. The **target utterance** $U_t$ is the $t^{th}$ utterances of a conversation $\mathcal{D} = (U_1, U_2, U_3, \ldots, U_N)$ where $N$ is the maximum number of utterances in this conversation and $0 < t \leqslant N$. The **emotion label** $Emo_t$ denotes the emotion type of $U_t$. The **emotion-cause pair (ECP)** is a pair $(U_t, U_i)$, where $U_i$ is the $i^{th}$ utterance of this conversation. In the ECP, $U_t$ represents the emotion utterance and $U_i$ is the corresponding cause utterance. Moreover, the **cause label** $C_{t,i}$ denotes the cause span type of the ECP $(U_t, U_i)$.

Thus, in a given text, **ERC** is the task of identifying all $Emo_t$. Moreover, **ECPE** aims to extract a set of ECPs and **ECSR** aims to identify all $C_{t,i}$.

### 2.2 Affective Reasoning in Conversation

Chen et al. (2018) introduced the pioneering work on ERC due to the growing availability of public conversational data. Building upon this, Xia and Ding (2019) further advanced the field by proposing the ECPE that jointly identifies both emotions and their corresponding causes. Moreover, Poria et al. (2021) has extended ECPE into conversations and proposed a novel ECSR task, specifically designed to identify ECP spans within conversation contexts. More recently, increasing works have indicated the crucial role played by accurate inference models in facilitating complex reasoning within these tasks, such as the assumption about interlocutors (Zhang et al., 2019; Lian et al., 2021; Shen et al., 2021) and context (Ghosal et al., 2019; Shen et al., 2022; Chen et al., 2023).

### 2.3 Challenge of Affective Reasoning

We examined the performance of a range of methods for addressing affective reasoning in conversations, including both unsupervised approaches (large language models (LLMs), BERT-based pretrained models) and supervised approaches (task-related approaches).

Overall, all the methods demonstrated a lack of discriminability on two types of challenges:

- Samples where emotional utterances and causal utterances are interchanged. For a dialogue instance, if the ECP is $(U_1, U_2)$ ($U_2$ is the cause of $U_1$), the prediction results obtained by the existing methods tend to include both $(U_1, U_2)$ and $(U_2, U_1)$.

| Methods | Challenge 1 | | Challenge 2 | | |
|---|---|---|---|---|---|
| | $(U_a, U_b)$ | $(U_b, U_a)$ | $(U_a, U_b)$ | $(U_b, U_c)$ | $(U_a, U_c)$ |
| GPT-3.5 | 112 | 102 | 108 | 114 | 109 |
| GPT-4 | 127 | 114 | 111 | 105 | 103 |
| RoBERTa | 95 | 97 | 94 | 91 | 83 |
| RoBERTa$^+$ | 97 | 91 | 105 | 101 | 106 |
| RANK-CP | 142 | 125 | 147 | 129 | 131 |
| ECPE-2D | 151 | 153 | 142 | 138 | 146 |
| EGAT | 166 | 154 | 157 | 139 | 148 |

Table 1: For the two challenges mentioned in Section 2.3, we conducted tests on a subset of 200 samples from the RECCON dataset. We recorded the number of samples identified by above methods. In the second row of Challenge 1, we showed the count of samples where these methods extracted the negative pairs in reverse cause order. Similarly, in the third row of Challenge 2, we showed the count of samples where these methods extracted negative indirect pairs.

- Samples with indirect connections. For example, if the ECPs in a conversation are $(U_1, U_2)$ and $(U_2, U_3)$, the prediction results obtained by the methods often include an additional pair $(U_1, U_3)$.

We evaluated the performance of existing methods on these two challenges, and the detailed results are shown in Table 1. All evaluated methods extracted a nearly equal number of negative samples as positive samples. Considering their performance in broad research domains, both unsupervised and supervised methods could demonstrate a desirable fitting ability to capture the semantic similarity between two utterances. This often apparently results in satisfactory performance in most tasks. However, when it comes to more specific causal relationships within semantically similar sentences (such as reasoning tasks), they may not exhibit the same level of "intelligence" and output some "pseudo-correlation".

In the area of causal discovery, Causal Markov and Faithfulness Assumptions (Spirtes et al., 2000; Colombo et al., 2012; Ogarrio et al., 2016), provide insights into capturing more specific causal relationships in the situation of the above challenges. Considering two similar variables: $A$ and $B$ that can be fitted, the independence tests enable the determination of specific causal relationships, such as "$A \rightarrow B$," "$B \rightarrow A$," or "$A \rightarrow L \rightarrow B$". More recently, the Structural Causal Model (SCM) (Shimizu et al., 2006; Shimizu and Bollen, 2014; Sanchez-Romero et al., 2019) built upon the independent noise assumptions has emerged as an effective approach to the limitation of Markov equivalence classes in distinguishing causal relationships.

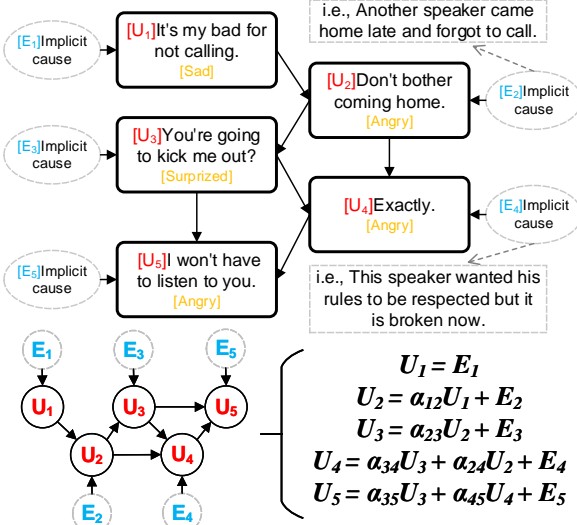

$$U_1 = E_1$$
$$U_2 = \alpha_{12}U_1 + E_2$$
$$U_3 = \alpha_{23}U_2 + E_3$$
$$U_4 = \alpha_{34}U_3 + \alpha_{24}U_2 + E_4$$
$$U_5 = \alpha_{35}U_3 + \alpha_{45}U_4 + E_5$$

Figure 1: The conversation case with five utterances. In the SCM, we assume that each utterance $U_i$ has a corresponding implicit cause $E_i$, and has several explicit causes. i.e., $U_4$ has an implicit cause $E_4$ and two explicit causes $U_3$ and $U_2$. In the lower part of the figure, SCM adopts $U_t = \sum \alpha_{it}U_i + E_t$ to denote these relationships and formalize the conversation as a DAG.

The noise terms (also called exogenous variables) for each variable, enables methods such as Independent Component Analysis (ICA) to identify more comprehensive causal relationships between the two fitted variables.

## 3 Methodology

In this section, we begin by outlining incorporating *i.i.d.* noise terms into a dialogue model to construct an SCM in Section 3.1, demonstrating independent residual allowing for the identification of more specific causal relations within pairs of fitted utterances. Next, to mitigate conflicts between SCM models and dialogue data, we designed *cogn* skeletons with six instantiations in Section 3.2. Finally, we propose a deep learning implementation to tackle the issue of noise being unknown in dialogue data in Section 3.3.

### 3.1 Structural Causal Model

In order to imbue causal discriminability into the fitting process of two relevant utterances, we algebraically construct the conversation model as a Structural Causal Model (SCM).

**Definition 1**: An SCM of a dialogue is a 3 tuple $\langle U, E, \mathcal{F} \rangle$, where $U$ is the set of utterances $U = \{U_i\}_{i=1}^N$, $E$ is the set of exogenous noises $E = \{E_i\}_{i=1}^N$ corresponding to each $U_i$, $N$ is

the number of utterances. Note that each $E_i$ is independent in the SCM. Structural equations $\mathcal{F} = \{f_i\}_{i=1}^N$ are functions that determine $U$ with $U_i = f_i(rel_{U_i}) + E_i$, where $rel_{U_t}$ denotes a set of utterances that point to the $U_t$.

Definition 1 establishes the construction of a novel computational model for dialogue process, as exemplified in Figure 1. In such a computational model, each utterance is endogenous and influenced by an independent exogenous variable. For simplicity, we refer to the variable $U$ as the explicit causes and the variable $E$ as the implicit causes. The independence of the implicit causes makes the residual terms meaningful during the fitting of any two utterances.

**Definition 2**: The relationship of two utterances $X$ and $Y$ in a dialogue is causal discriminable, from the independent conditions:

- $\Sigma_X \perp\!\!\!\perp Y, \Sigma_Y \not\!\perp\!\!\!\perp X \Rightarrow Y \rightarrow X$

- $\Sigma_X \not\!\perp\!\!\!\perp Y, \Sigma_Y \perp\!\!\!\perp X \Rightarrow X \rightarrow Y$

- $\Sigma_X \not\!\perp\!\!\!\perp Y, \Sigma_Y \not\!\perp\!\!\!\perp X \Rightarrow L \rightarrow X, L \rightarrow Y$

- $\Sigma_X \perp\!\!\!\perp Y, \Sigma_Y \perp\!\!\!\perp X \Rightarrow X \rightarrow L, Y \rightarrow L$

where $\Sigma$ represents the residual terms in fitting process. (The proof is shown in Appendix A.)

**Example 1**: A 4-utterance dialogue SCM $D = \{\{U_1, U_2, U_3, U_4\}, \{E_1, E_2, E_3, E_4\}, \{a, b, c\}\}$ with the true relationships are $U_1 = E_1$, $U_2 = aU_1 + E_2$, $U_3 = bU_1 + E_3$, $U_4 = cU_3 + E_4$. The fitting of $U_2$ with $U_3$ and $U_4$ yield $U_2 = \frac{a}{b}U_3 + 0U_4 + \Sigma_{U_2}$, while the fitting of $U_3$ with $U_2$ and $U_4$ yield $U_3 = \frac{b}{a}U_2 + \frac{1}{c}U_4 + \Sigma_{U_3}$. Additionally, the fitting of $U_4$ with $U_2$ and $U_3$ lead to $U_4 = 0U_2 + cU_3 + \Sigma_{U_4}$.

In Example 1, it is observed that any two utterances can be fitted together as they are mutually dependent. However, causal discriminability can be employed to differentiate their distinct causal structures. For instance, the residual term $\Sigma_{U_3}$ is not independent of $U_4$, while $\Sigma_{U_4}$ is independent of $U_3$, indicating that $U_3$ is a cause of $U_4$. Furthermore, the residual term $\Sigma_{U_3}$ is not independent of $\Sigma_{U_2}$, and $\Sigma_{U_2}$ is not independent of $U_3$, implying the presence of common cause ($U_1$) between $U_2$ and $U_3$.

## 3.2 Causal Skeleton Estimation

Establishing a skeleton is the first step in causal discovery, as different skeletons provide distinct learning strategies for recovering the relationships between variables. However, utterances differ from the variables that causal discovery often uses. Specifically, each conversation has a different amount ($N$) of utterances, and different inter-utterances relationships related to the context. Hence, it is intractable to build a general causal skeleton with fixed nodes and edges to describe all conversation samples.

Fortunately, several published GNN-based approaches (Shen et al., 2021; Ishiwatari et al., 2021; Ghosal et al., 2019; Lian et al., 2021; Zhang et al., 2019) in ERC have proposed and verified a common hypothesis to settle down this issue. The Hypotheses are elaborated on in Appendix B.

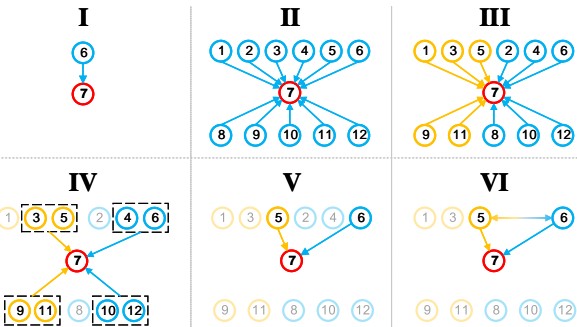

Figure 2: Six *cogn* skeletons from a conversation case with 12 utterances. We adopted the 7-th utterance as the target utterance (Red). Orange nodes denote the utterances of the same speaker as the target utterance, and blue ones denote those belonging to other speakers. Arrow represents the information propagated from one utterance to another, and the bi-way arrow represents the influence-agnostic relationship. The black dash box represents the slide windows.

Figure 2 showcases the design of six *cogn* skeletons, derived from the latest works that have employed one or more of these hypotheses. The statistic and specific algorithms are also shown in Appendix B. Note that we only conduct experiments for **II-VI** because our structure is hard to apply with the recurrent-based skeleton.

## 3.3 Approach Implementation

From a given causal skeleton, a linear SCM can be equivalently represented as:

$$U_t = \sum_{i \in rel_t} \alpha_{i,t} U_i + E_t \qquad (1)$$

where $rel_t$ denotes a set of utterances that point to the $U_t$ (7-th utterance) in Figure 2, $E_t$ represents the exogenous variable towards the variable $U_t$ in

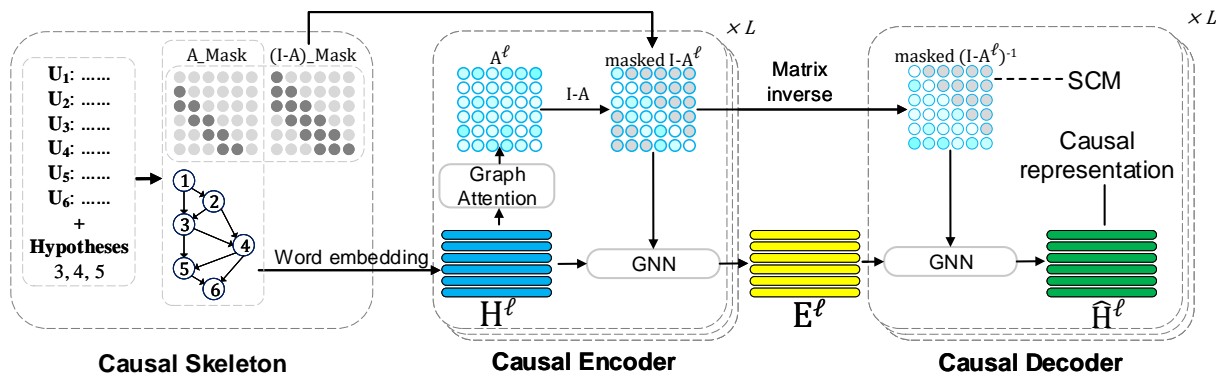

Figure 3: Processing of our approaches, with a six-utterances conversation case as the input. Causal skeleton indicates which utterances (nodes) are used for aggregation. For each layer $\ell$, we collect representations $H^\ell$ for all utterances where each row represents one utterance. Causal Encoder yields the implicit causes $E^\ell$, the input for Decoder learning the causal representation. In all matrices, light gray nodes represent the masked part.

SCM, as well as the implicit cause towards the utterance $U_t$ in conversation. Furthermore, we denote the word embedding of $U$ by $H = h_1, h_2, \ldots, h_N$, and the relationships between utterances in rows can also be written as: $H = A^T H + E$, where $A_{i,t} \neq 0$ stands for a directed edge from $U_i$ to $U_t$ in SCM. Thus we can define the Graph $\mathcal{G} = (\mathcal{V}, \mathcal{E})$ with adjacency matrix $A_{i,i} = 0$ for all $i$.

However, in this equation, only $H$ is known. The unknown variable $A$ is typically the target of inference, while the unknown variable $E$ represents exogenous variables that implicitly influence each utterance, such as the speaker's memory, experiences, or desires. (Krych-Appelbaum et al., 2007; Sidera et al., 2018) These factors are typically missing in existing conversation resources. Therefore, determining $A$ based on completely unknown $E$ is another problem we aim to address.

Hence, we treat $A^T$ as an autoregression matrix of the $\mathcal{G}$, and then $E$ can be yielded by an auto-encoder model. The whole process reads:

$$E = f((I - A^T)H) \qquad (2)$$

$$\widehat{H} = g((I - A^T)^{-1}E) \qquad (3)$$

where $f(\cdot)$ and $g(\cdot)$ represent the encoder and decoder neural networks respectively. Encoder aims to generate an implicit cause $E$, and Decoder devotes to yielding a causal representation $\widehat{H}$. From Equation 1, causal representation $\widehat{H}_t$ reasons about the fusion relations of heterogeneous explicit causes $\sum_{i \in rel_t} H_i$ and implicit cause $E_t$. The details of this process are shown in Figure 3.

**Encoder.** We use the graph attention mechanism to learn the adjacency matrix $A$ and construct a hierarchical GNN to instantiate the $f(\cdot)$.

$\ell = 1, 2, \ldots, L - 1$ represents the layer of GNN. Thus, for each utterance at the $\ell$-th layer, the $A_{i,t}^\ell$ computed by attention mechanism is a weighted combination of $h_t^\ell$ for each directly related utterance $U_i (i \in rel_t)$:

$$A_{i,t}^\ell = \frac{LeakyReLU(e_{i,t}^\ell)}{\sum_{j \in rel_t} LeakyReLU(e_{j,t}^\ell)} \qquad (4)$$

$$e_{i,t}^\ell = \overrightarrow{h}_i W_{i(row)}^\ell + (\overrightarrow{h}_t W_{t(col)}^\ell)^T \qquad (5)$$

where $W_{row}^\ell \in \mathbb{R}^{N \times 1}$ and $W_{col}^\ell \in \mathbb{R}^{N \times 1}$ are the learnable parameters in the graph attention. Moreover, the GNN aggregates the information from the neighbor utterances as following:

$$H^{\ell+1} = eLU((I - (A^\ell)^T)H^\ell W^\ell) \qquad (6)$$

where $W^\ell$ stands for parameters in the corresponding layer. From the final layer of the evaluation process, by extracting $A^{L-1}$ computed in Equation 4, the marginal or conditional "distribution" of $H$ is obtained, showing how to discover Causal Graph $\mathcal{G}$ from $\mathcal{D}$. Besides, by extracting $H^L$ in Equation 6, we can obtain the independent embedding for the implicit causes $E = MLP(H^L)$.

**Decoder.** We utilize the $A$ and $E$ computed from Encoder to generate the causal representation $\widehat{H}$. With a fixed adjacency matrix $A$, the GNN aggregates the information of implicit causes from neighbor nodes as follows:

$$\widehat{E}^{\ell+1} = eLU((I - (A^L)^T)^{-1}E^\ell M^\ell) \qquad (7)$$

where $M^\ell$ is parameters in the corresponding layer. As the same architecture as the encoder, $\widehat{H} =$

$MLP(E^L)$. Additionally, the plug-in RNN is integrated with GNN to address the appetite of **Hypothesis 6**:

$$\widehat{E}^{\ell+1} = GRU^\ell(\widehat{E}^\ell, p^\ell) \qquad (8)$$

where $p^\ell$ is the state of GRU model, with $p$ computed by self-attention proposed by Thost and Chen (2021).

### 3.4 Optimization

In our approach, $\widehat{H}$ and $H$ acts identically under the **linear** SCM model. Similarly, $\widehat{H}$ should be aligned with $H$ in emotion dimensions under the **non-linear** SCM model. In short, we adopt an auxiliary loss measuring the Kullback-Leibler (KL) divergence (Joyce, 2011) of $\widehat{H}$ and $H$ mapped into the exact emotion dimensions. Moreover, implicit causes $E$ is one of the crucial influence factors on $\widehat{H}$, so that the loss aims to impose the constraint that $\widehat{H}$ is the embedding of our need to ensure generating correct $E$.

$$Loss_{KL} = \sum_t \sum_{e \in Emo_t} p_e(\widehat{U_t}) \log \frac{p_e(\widehat{U_t})}{p_e(U_t)} \qquad (9)$$

where $e$ is any emotion type in $Emo_t$, $p_e$ denotes the probability labeled with emotion $e$. In the whole process of ARC tasks, we followed (Wei et al., 2020; Poria et al., 2021) to add several losses of ECPE and ECSR respectively.

Furthermore, we would like to explain the difference between our approach and Variational Auto-Encoder (VAE) (Kingma and Welling, 2014). The output of the encoder in VAE is $q_\phi(Z)$. With this estimation of $\widehat{q}_\phi(Z)$, we can measure the variation $\xi(q_\phi(Z))$ (also called $\nabla_\phi ELBO(\widehat{q}_\phi(Z))$) to obtain the approximation estimation of $ELBO(q)$. In contrast, our output is $E$, a fixed matrix rather than a distribution. In other words, the VAE depends on the prior distribution over the latent variables, whereas our approach has a dependency on the consistency of $H$ and $\widehat{H}$, which is non-sampling and non-distributive.

## 4 Experiments

In this section, we conduct extensive experiments to answer the 3 research questions:

**RQ1:** How effective is our method in affective reasoning tasks?

**RQ2:** How do we justify the causal discriminability of our method?

**RQ3:** How do we gauge the difference between the latent variable $E$ and designed implicit causes?

| Dataset | conversations | | | tasks | | |
|---|---|---|---|---|---|---|
| | Train | Val | Test | ERC | ECPE | ECSR |
| DailyDialog | 11118 | 1000 | 1000 | √ | – | – |
| MELD | 1038 | 114 | 280 | √ | – | – |
| EmoryNLP | 713 | 99 | 85 | √ | – | – |
| IEMOCAP | 100 | 20 | 31 | √ | – | – |
| RECCON-DD | 833 | 47 | 225 | – | √ | √ |
| RECCON-IE | – | – | 16 | – | √ | √ |
| Synthetic data | 833 | 47 | 225 | – | √ | √ |

Table 2: The statistics of seven datasets

### 4.1 Datasets, Implementation and Baselines

We use six real datasets for three affective reasoning tasks and one synthetic dataset for justifying $E$ in our model. The statistics of them are shown in Table 2. Appendix C depicts the detailed introductions of each dataset.

We adopt the consistent benchmarks of the SOTA methods in three tasks, including the pre-training language model, hyper-parameters, $t$-tests, and metrics. The implementation details are shown in Appendix D.

According to the hypotheses of these baselines, for each *cogn* skeleton, we choose one recent SOTA work: II: **DialogXL** (Shen et al., 2022). III: **EGAT** (Chen et al., 2023). IV: **RGAT** (Ishiwatari et al., 2021). V: **DECN** (Lian et al., 2021). VI: **DAG-ERC** (Shen et al., 2021).

### 4.2 Overall Performance (RQ1)

Table 3 reports the results of ECPE and ECSR, with $p < 0.01$ in the $t$-test, where the best improvement and best performance both concentrate on **VI**. With the visualization of Appendix F, we infer that the upper triangular adjacency matrix of DAG-ERC, not restricted by the backpropagation, benefits from **Hypothesis 6**. Moreover, **II** lags farthest behind in the ECPE while achieving the second best in the ECSR, showing that the reliance on a hypothesis is not equal in different tasks. Furthermore, without **Hypotheses 1** and **6**, **III**, **IV**, and **V** are far from the best performance since **Hypothesis 1** has the maximum identifying space, and **Hypothesis 6** supplies the highest number of information passing. Finally, it is worth noting that three skeleton-agnostic baselines and unsupervised methods perform poorly in the RECCON-IE dataset, indicating that our models have stronger representation learning capabilities as well as highlighting the continued research value of affective reasoning tasks.

We further conducted six sets of ablation experiments to study the effects of different modules. In Table 4, we summarized results under the following cases: replacing $Loss_{KL}$ with $BCE$ loss function

| Skt | model | ECPE in RECCON | | ECSR in RECCON | |
|---|---|---|---|---|---|
| | | DD($\pm\sigma_{10}$) | IE | DD($\pm\sigma_{10}$) | IE |
| – | GPT-3.5 | 38.13 | 39.55 | 10.49 | 5.36 |
| | GPT-4 | 46.29 | 49.32 | 16.81 | 18.39 |
| | RoBERTa | 53.91±1.5 | 38.77 | 31.52±0.8 | 20.16 |
| | RoBERTa$^+$ | 54.62±1.1 | 38.26 | 33.28±0.7 | 26.37 |
| | RANK-CP† | 63.51±2.1 | 41.56 | 26.57±0.8 | 18.99 |
| | ECPE-2D† | 64.35±1.7 | 47.42 | 34.41±0.1 | 22.03 |
| II | DialogXL† | 61.92±1.7 | 50.31 | **35.79±0.5** | 21.78 |
| | +Ours | **64.74±1.6** | **51.23** | 34.63±0.2 | **27.92** |
| III | EGAT | 68.05±1.5 | 53.43 | 29.68±0.7 | 16.42 |
| | +Ours | **69.16±1.2** | **53.81** | **30.5±0.2** | **18.55** |
| IV | RGAT† | 69.02±1.9 | 52.48 | **30.39±0.4** | 17.49 |
| | +Ours | **70.12±2.1** | **53.93** | 30.24±0.5 | **19.31** |
| V | DECN† | 68.32±1.5 | 51.73 | 30.7±0.9 | 18.47 |
| | +Ours | **68.84±1.7** | **53.89** | **31.88±0.2** | **20.13** |
| VI | DAG-ERC† | 70.36±1.5 | 55.7 | 40.12±0.7 | 24.89 |
| | +Ours | **73.17±1.1** | **56.67** | **42.14±0.1** | **30.41** |

Table 3: Overall performance in ECPE and ECSR tasks. We additionally compare four unsupervised approaches and two baselines not belonging to any skeleton: RANK-CP (Wei et al., 2020), ECPE-2D (Ding et al., 2020). The RoBERTa$^+$ represents the large RoBERTa version (1024 dimensions). The DD and IE are two subsets (see Appendix C).

| Model | Categories | | | | |
|---|---|---|---|---|---|
| | **II** | **III** | **IV** | **V** | **VI** |
| Ours | 64.74 | 69.16 | 70.12 | 68.84 | 73.17 |
| $BCE$ | ↓0.62 | ↓0.04 | ↓0.15 | ↓0.16 | ↓0.29 |
| w/o $Loss_{KL}$ | ↓2.18 | ↓1.95 | ↓**2.42** | ↓1.33 | ↓1.58 |
| w/o Decoder | ↓**3.59** | ↓**2.79** | ↓2.11 | ↓**2.83** | ↓**4.14** |
| w/o Hypo 6 | – | – | – | – | ↓1.59 |
| w/o Hypo 5 | – | – | – | ↓2.34 | ↓1.88 |
| w/o Hypo 4 | – | ↓**3.67** | ↓**2.72** | ↓**3.15** | ↓**4.19** |

Table 4: Ablation results

| Methods | Reversal | | Chain | | Common Cause | |
|---|---|---|---|---|---|---|
| | Pos | Neg | Pos | Neg | Pos | Neg |
| GPT-3.5 | 45.9 | 41.2 | 43.4 | 44.7 | 41.6 | 36.4 |
| GPT-4 | 49.3 | 46.2 | 48.9 | 43.7 | 47.7 | 48.1 |
| RoBERTa | 53.9 | 52 | 56.4 | 53.1 | 59.3 | 56.7 |
| RoBERTa$^+$ | 56.7 | 54.9 | 58.7 | 56.1 | 52.6 | 54.6 |
| RANK-CP | 61.7 | 62.5 | 63.4 | 61.5 | 65.9 | 63.7 |
| ECPE-2D | 63.9 | 62.8 | 64.6 | 61.3 | 63.3 | 61.9 |
| DialogXL | 64.8 | 60.8 | 61.9 | 63.8 | 65 | 66.1 |
| EGAT | 68.7 | 64.3 | 68.8 | 64.8 | 66.2 | 64.3 |
| RGAT | 69.4 | 61.7 | 68.9 | 66.4 | 68.3 | 68.5 |
| DECN | 66.7 | 62.4 | 70.5 | 64.3 | 69.2 | 66.1 |
| DAG-ERC | 71.5 | 68.2 | 72.4 | 64.2 | 69.3 | 66.1 |
| Ours | 76.2 | 46.1 | 73.8 | 41.9 | 77.2 | 48.6 |

Table 5: Results of causal discriminability. The calculation results = count of predicted results that matched the samples / total number of samples *100. The three causal models are: Reversal Model: Positive samples (i, j) and negative samples (j, i); Chain Model: Positive samples (i, k) and (k, j), and negative samples (i, j); Common Cause Model: Positive samples (k, i) and (k, j), and negative samples (i, j) or (j, i). The metric "Pos" represents the percentage of positive samples, indicating the extraction capability. Higher Pos samples imply a better extraction capability. The metric "Neg" represents the percentage of negative samples. A smaller difference between Pos and Neg indicates a weaker causal discriminability.

Furthermore, Appendix E reports the results of ERC task and sensitivity experiments to analyze how our model performs in different $L$ and $k$.

### 4.3 Relationship analysis (RQ2)

We are also concerned about the causal discriminability for similar utterances. Table 5 demonstrates that in all three different causal models, none of the methods were able to distinguish between negative and positive samples. Because both negative and positive samples can be fit within these three causal models, solely from an embedding similarity perspective. However, our method significantly decreases the percentage of negative samples indicating the effectiveness of incorporating implicit cause noise to enhance causal discriminative ability.

Additionally, we show the adjacent matrices of our model and current SOTA methods in Appendix F. which indicates that our model can more freely explore the relationship between different utterances via adjacent matrices shifting rather than being limited to a fixed structure (e.g., attention module).

($BCE$); removing the $Loss_{KL}$ (w/o $Loss_{KL}$); replacing Decoder module with a Linear layer (w/o Decoder); removing the RNN module (w/o **Hypo 6**); adding the edges from successors to predecessors (w/o **Hypo 5**); reducing the speaker types to one (w/o **Hypo 4**).

As shown in Table 4, $BCEloss$ performs similarly to $Loss_{KL}$; thus, we empirically demonstrate that our auxiliary loss is essentially different from $Loss_{KL}$ in VAE. The F1 score decreases heavily without auxiliary loss or decoder, these two are necessary ingredients for building complete processing to learn the causal representation via $E$. Besides, **Hypotheses 4, 5**, and **6** are all critical but removing **Hypothesis 4** leads to the highlight degradation in 3 skeletons. This result corroborates the theory of Lian et al. (2021) and Shen et al. (2021), who state that speaker identity is the strong inductive bias in conversation. Finally, it is expected to see that skeleton with **Hypotheses 4, 5, and 6** should be the closest to perfection while the DAG-ERC+Ours indeed achieves the SOTA.

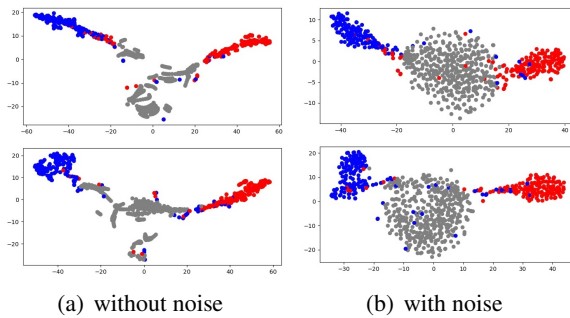

(a) without noise  (b) with noise

Figure 4: Visualization of $E$ (upper subfigures) and implicit causes (lower subfigures) with colors in the simulated datasets. The gray cluster means padding utterances in each dialogue, the blue cluster corresponds to the non-emotion utterances, and the red cluster corresponds to emotion utterances.

### 4.4 Implicit Causes Study (RQ3)

The latent variable $E$ is intended to represent the mentioned implicit causes. Therefore, the global distribution of the latent variable $E$ should be approximately equal to the one of implicit causes. Although human evaluation labels are better for proving reasonable performance, it is intractable to annotate implicit causes due to their unobservability. We thus trained our model in a synthetic dataset given a set of fixed *i.i.d.* implicit causes to observe how the $E$ is similar to the ground truth implicit causes distributions. Figure 4 (a-b) shows the projection of $E$ and implicit causes, respectively, using t-SNE (Knyazev et al., 2019). We observe that $E$ and implicit causes are both similarly clustered into three parts through the distribution properties. $E$ is consistent with the implicit causes in the samples with or without noise indicating that $E$ successfully learns the implicit causes.

Moreover, in Appendix G, we first prove the approximate emotion consistency between utterance $U_t$ and its implicit causes when $U_t$ and $U_i$ in the emotion-cause pair $(U_t, U_i)$ do not belong to the same emotion category. Then, we demonstrate through the ERC task that by replacing $\hat{H}$ with $E$, the emotion consistency provided by implicit causes is preserved.

### 4.5 Limitations

In our model, our method can distinguish between $U_i \rightarrow U_j$ and $U_i \leftarrow U_k \rightarrow U_j$. However, our method is unable to distinguish between $U_i \rightarrow U_j$ and $U_i \leftarrow L \rightarrow U_j$, where L represents a unobserved variable, called common causes or con-

founders. In Tables 3, 7, and 8, skeletons **II**, **III**, and **IV** generally lag far behind **V** and **VI**. This unsatisfactory performance of these skeletons indicates that excessive adding-edge leads to serious confounders.

Therefore, we proposed a theoretical design for testifying the existing of latent confounders:

**Confounding between Non-adjacent Nodes**: Consider two utterances $U_i$ and $U_j$ being non-adjacent utterances. Let $Pa$ be the union of the parents of $U_i$ and $U_j$: $Pa = U_i \cup U_j$. If we perform an intervention on $Pa$ (i.e., $do(Pa = pa)$), we thus have $U_i \perp\!\!\!\perp U_j$ if and only if there is a latent confounder $L$ such that $U_i \leftarrow L \rightarrow U_j$.

**Confounding between Adjacent Nodes**: Consider two utterances $U_i$ and $U_j$ being adjacent utterances: $U_i \rightarrow U_j$. If there are no latent confounders, we have $P(U_j|U_i) = P(U_j|do(U_i = u_i))$.

Indeed, implementing intervention operations on conversation data poses a significant challenge. Therefore, in our new work, we have proposed general intervention writing: $do(X) := Pa(X) = \emptyset$ where $Pa(X)$ denotes the parent set. Moreover, the most significant obstacle to further research is the lack of a high-quality dataset with complete causal relationship labels. Hence, we have constructed a simulated dialogue dataset via GPT-4 and plan to make it open soon.

## 5 Conclusion

The results of testing prevalent approaches on the ARC task have demonstrated that almost all approaches are unable to determine the specific causal relationship that leads to the association of two well-fitted embeddings. In order to enhance the causal discrimination of existing methods, we constructed a SCM with *i.i.d.* noise terms, and analyzed the independent conditions that can identify the causal relationships between two fitted utterances. Moreover, we proposed the *cogn* framework to address the unstructured nature of conversation data, designed an autoencoder implementation to make implicit cause learnable, and created a synthetic dataset with noise labels for comprehensive experimental evaluation. While our method still has some limitations, such as confounders and the inability to scale to all methods, we hope that our theory, design, and model can provide valuable insights for the broader exploration of this problem to demonstrate that our work is *de facto* need for identifying causal relationships.

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

## A Proof of Definition 2

Let $X$ and $Y$ be two variables in an SCM, with their respective noise terms denoted as $E_X$ and $E_Y$ (where $E_X$ and $E_Y$ are mutually independent). Let $\hat{X}$ and $\hat{Y}$ represent the fitted values of $X$ and $Y$ w.r.t. each other: $\hat{X} = \lambda Y$ and $\hat{Y} = \frac{1}{\lambda}X$. The residual terms between the fitted values and the true values are denoted as $\Sigma_X = X - \hat{X}$ and $\Sigma_Y = Y - \hat{Y}$. The true strength of $Y \to X$ is $k$.

Hence, if the SCM only contains two variables writing:

$$Y = E_Y \tag{10}$$

$$X = kY + E_X \tag{11}$$

The residual terms could write:

$$\Sigma_X = X - \lambda(\frac{1}{k}(X - E_X)) \tag{12}$$

$$\Sigma_Y = Y - \frac{1}{\lambda}(ky + E_X) \tag{13}$$

Then, if the true causal relationship is from $Y$ to $X$, $\lambda = k$. $\Sigma_X$ does not contain the term of $E_Y$ while $\Sigma_Y$ contains the term of $E_X$. We could obtain the independence of residual terms writting:

$$\Sigma_X = \lambda E_X \perp\!\!\!\perp Y \tag{14}$$

$$\Sigma_Y = \frac{1}{\lambda}E_X \not\perp\!\!\!\perp X \tag{15}$$

and vice versa. Therefore, we could obtain the independence condition:

- $\Sigma_X \perp\!\!\!\perp Y, \Sigma_Y \not\perp\!\!\!\perp X \Rightarrow Y \to X$

- $\Sigma_X \not\perp\!\!\!\perp Y, \Sigma_Y \perp\!\!\!\perp X \Rightarrow X \to Y$

Furthermore, there may exist a set of independence: $\Sigma_X \not\perp\!\!\!\perp Y, \Sigma_Y \not\perp\!\!\!\perp X$. We would like to assume that there is a latent variable $L$, for this situation, constructing two relationships $L \to X$ and $L \to Y$. Then we obtain: $\Sigma_L \not\perp\!\!\!\perp X, \Sigma_L \not\perp\!\!\!\perp Y$. By utilizing the transitivity of conditional independence, we can establish $X \not\perp\!\!\!\perp Y$, and finally acheive the situation $\Sigma_X \not\perp\!\!\!\perp Y, \Sigma_Y \not\perp\!\!\!\perp X$. We likewise assume a latent variable $L$ establishing $X \to L$ and $Y \to L$ for the opposite situation where $\Sigma_X \perp\!\!\!\perp Y, \Sigma_Y \perp\!\!\!\perp X$, and $X, Y$ are two isolated variables in SCM. From the above independence conditions, we could obtain: $\Sigma_L \perp\!\!\!\perp X, \Sigma_L \perp\!\!\!\perp Y$. Due to the graph structure of SCM, we could obtain: $\Sigma_X \perp\!\!\!\perp Y, \Sigma_Y \perp\!\!\!\perp X \Rightarrow X \perp\!\!\!\perp Y$. Considering the residual terms, we finally obtain: $X \not\perp\!\!\!\perp \Sigma_X$ and $X \perp\!\!\!\perp Y \Rightarrow \Sigma_X \perp\!\!\!\perp Y$ and $Y \not\perp\!\!\!\perp \Sigma_Y$ and $Y \perp\!\!\!\perp X \Rightarrow \Sigma_Y \perp\!\!\!\perp X$.

Hence, we could obtain additional two independence conditions:

- $\Sigma_X \not\perp\!\!\!\perp Y, \Sigma_Y \not\perp\!\!\!\perp X \Rightarrow L \to X, L \to Y$

- $\Sigma_X \perp\!\!\!\perp Y, \Sigma_Y \perp\!\!\!\perp X \Rightarrow X \to L, Y \to L$

Based on the independence conditions of 2-variables SCM, we could extend it to the general SCM including more than 2 variables. Given any two variables in a SCM, we could testify to the independence condition and finally orientate via the whole SCM.

## B Hypotheses and Algorithms for Skeletons

**Hypothesis 0.** $\forall U_i \in \mathcal{D}$, *it has the same causal skeleton as other utterances.*

By regarding **Hypothesis 0** as the prior knowledge, a common causal skeleton containing a target variable and a fixed number of related variables can reason about the relations between the target utterance and other considered utterances. We denote this skeleton of $U_t$ by $S(U_t)$. There are $\forall U_i, U_j \in \mathcal{D}, S(U_i) = S(U_j)$.

Additionally, there are some other empirical hypotheses from the above approaches. These hypotheses can be divided into two categories: one is about the "order" of utterances (**Hypotheses 1, 2, 3**), and the other is about intermingling dynamics among the interlocutors (**Hypotheses 4, 5, 6**).

**Hypothesis 1.** (Majumder et al., 2019) *Under the sequential order, the target utterance receives information only from the previous utterance.*

| Category | Hypothesis | Original work |
|----------|------------|---------------|
| I | 1 | Majumder et al. (2019) |
| II | 2 | Veličković et al. (2017) |
| III | 2,4 | Chen et al. (2023) |
| IV | 3, 4 | Ghosal et al. (2019) |
| V | $3(k = 1)$, 4, 5 | Lian et al. (2021) |
| VI | 3, 4, 5, 6 | Shen et al. (2021) |

Table 6: Statistics of 6 *cogn* skeletons. We detailed the hypotheses each *cogn* skeleton adopted and the original works from which we designed them.

**Hypothesis 2.** (Wei et al., 2020) *Under the graph order, the target utterance receives information from all other utterances.*

**Hypothesis 3.** (Ghosal et al., 2019) *Under the local graph order, target utterance receives local information from k surround utterances.*

**Hypothesis 4.** (Zhang et al., 2019) *The influence between two utterances can be discriminated by whether the two utterances belong to the same speaker identity.*

**Hypothesis 5.** (Lian et al., 2021) *Target utterance only receives information from the predecessor utterances.*

**Hypothesis 6.** (Shen et al., 2021) *Between two utterances both related to the target utterance, there is also information passing, often dubbed as a partial order.*

A *cogn* skeleton is denoted by $\mathcal{H} = (\mathcal{V}, \mathcal{E}, \mathcal{M})$. The $\mathcal{V} = U_1, U_2, U_3, ..., U_N$ represents a set of utterances in a conversation, and the edge $(i, j, m_{i,j}) \in \mathcal{E}$ denotes the influence from $U_i$ to $U_j$, where $m_{i,j} \in \mathcal{M}$ is the type of the edge depending on whether $U_i$ and $U_j$ belong to one and the same speaker. Thus $\mathcal{M} = 0, 1$, where 1 for that they are the same speaker and 0 for different. Then we denote the speaker type of $U_i$ by a function $p(U_i)$. At last, we show the process of building 6 *cogn* skeletons in Algorithms $1 - 6$.

---

**Algorithm 1:** Buliding **I** *cogn* skeleton

**Input:** $\mathcal{D}, p(\cdot), k$
**Output:** $\mathcal{H} = (\mathcal{V}, \mathcal{E})$
1 $\mathcal{V} \leftarrow U_1, U_2, U_3, ..., U_N$
2 $\mathcal{E} \leftarrow \emptyset$
3 **forall** $i \in 2, 3, \ldots, N - 1$ **do**
4 $\quad \mathcal{E} \leftarrow \mathcal{E} \cup (i, i + 1)$
5 **return** $\mathcal{H} = (\mathcal{V}, \mathcal{E})$

---

**Algorithm 2:** Buliding **II** *cogn* skeleton

**Input:** $\mathcal{D}, p(\cdot), k$
**Output:** $\mathcal{H} = (\mathcal{V}, \mathcal{E})$
1 $\mathcal{V} \leftarrow U_1, U_2, U_3, ..., U_N$
2 $\mathcal{E} \leftarrow \emptyset$
3 **forall** $i \in 2, 3, \ldots, N$ **do**
4 $\quad$ **forall** $j \in 2, 3, \ldots, N$ **do**
5 $\quad\quad$ **if** $i! = j$ **then**
6 $\quad\quad\quad \mathcal{E} \leftarrow \mathcal{E} \cup (j, i)$
7 $\quad\quad$ **else**
8 $\quad\quad\quad Continue$
9 **return** $\mathcal{H} = (\mathcal{V}, \mathcal{E})$

---

**Algorithm 3:** Buliding **III** *cogn* skeleton

**Input:** $\mathcal{D}, p(\cdot), k$
**Output:** $\mathcal{H} = (\mathcal{V}, \mathcal{E}, \mathcal{M})$
1 $\mathcal{V} \leftarrow U_1, U_2, U_3, ..., U_N$
2 $\mathcal{E} \leftarrow \emptyset$
3 $\mathcal{M} \leftarrow 0, 1$
4 **forall** $i \in 2, 3, \ldots, N$ **do**
5 $\quad$ **forall** $j \in 2, 3, \ldots, N$ **do**
6 $\quad\quad$ **if** $p(U_j) = p(U_i)$ *and* $i! = j$ **then**
7 $\quad\quad\quad \mathcal{E} \leftarrow \mathcal{E} \cup (j, i, 1)$
8 $\quad\quad$ **else if** $p(U_j)! = p(U_i)$ *and* $i! = j$ **then**
9 $\quad\quad\quad \mathcal{E} \leftarrow \mathcal{E} \cup (j, i, 0)$
10 $\quad\quad$ **else**
11 $\quad\quad\quad Continue$
12 **return** $\mathcal{H} = (\mathcal{V}, \mathcal{E}, \mathcal{M})$

---

Finally, in Figure 5, we show the adjacency matrix of each *cogn* skeleton by inputting a binary alternating conversation case with 6 utterances. But

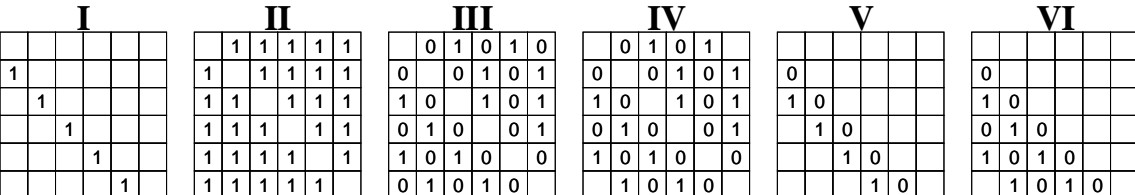

Figure 5: Adjacency matrices towards 6 *cogn* skeletons when $k = 2$. $(i, j) \neq None$ represents that $U_i$ is influenced by $U_j$.

---

**Algorithm 4:** Buliding **IV** *cogn* skeleton

**Input:** $\mathcal{D}, p(\cdot), k$
**Output:** $\mathcal{H} = (\mathcal{V}, \mathcal{E}, \mathcal{M})$
1   $\mathcal{V} \leftarrow U_1, U_2, U_3, ..., U_N$
2   $\mathcal{E} \leftarrow \emptyset$
3   $\mathcal{M} \leftarrow 0, 1$
4   **forall** $i \in 2, 3, \ldots, N$ **do**
5     **forall** $j \in 2, 3, \ldots, N$ **do**
6       **if** $p(U_j) = p(U_i)$ *and* $0 < |i - j| < k$ **then**
7         $\mathcal{E} \leftarrow \mathcal{E} \cup (j, i, 1)$
8       **else if** $p(U_j)! = p(U_i)$ *and* $0 < |i - j| < k$ **then**
9         $\mathcal{E} \leftarrow \mathcal{E} \cup (j, i, 0)$
10       **else**
11         *Continue*
12   **return** $\mathcal{H} = (\mathcal{V}, \mathcal{E}, \mathcal{M})$

---

**Algorithm 5:** Buliding **V** *cogn* skeleton

**Input:** $\mathcal{D}, p(\cdot), k$
**Output:** $\mathcal{H} = (\mathcal{V}, \mathcal{E}, \mathcal{M})$
1   $\mathcal{V} \leftarrow U_1, U_2, U_3, ..., U_N$
2   $\mathcal{E} \leftarrow \emptyset$
3   $\mathcal{M} \leftarrow 0, 1$
4   **forall** $i \in 2, 3, \ldots, N$ **do**
5     $\gamma \leftarrow i - 1$
6     **if** $p(U_\gamma) = p(U_i)$ **then**
7       $\mathcal{E} \leftarrow \mathcal{E} \cup (\gamma, i, 1)$
8     **else**
9       $\mathcal{E} \leftarrow \mathcal{E} \cup (\gamma, i, 0)$
10     $\gamma \leftarrow \gamma - 1$
11   **return** $\mathcal{H} = (\mathcal{V}, \mathcal{E}, \mathcal{M})$

---

**Algorithm 6:** Buliding **VI** *cogn* skeleton

**Input:** $\mathcal{D}, p(\cdot), k$
**Output:** $\mathcal{H} = (\mathcal{V}, \mathcal{E}, \mathcal{M})$
1   $\mathcal{V} \leftarrow U_1, U_2, U_3, ..., U_N$
2   $\mathcal{E} \leftarrow \emptyset$
3   $\mathcal{M} \leftarrow 0, 1$
4   **forall** $i \in 2, 3, \ldots, N$ **do**
5     $c \leftarrow 0$
6     $\gamma \leftarrow i - 1$
7     **while** $\gamma > 0$ *and* $c < k$ **do**
8       **if** $p(U_\gamma) = p(U_i)$ **then**
9         $\mathcal{E} \leftarrow \mathcal{E} \cup (\gamma, i, 1)$
10         $c \leftarrow c + 1$
11       **else**
12         $\mathcal{E} \leftarrow \mathcal{E} \cup (\gamma, i, 0)$
13       $\gamma \leftarrow \gamma - 1$
14   **return** $\mathcal{H} = (\mathcal{V}, \mathcal{E}, \mathcal{M})$

note that adjacency can not indicate all the differences among these skeletons, for example, Hypothesis 6 takes effect when the model learns the relationship based on the **VI** skeleton.

## C Datasets

**DailyDialog** (Li et al., 2017): A Human-written dialogs dataset with 7 emotion labels (*neutral, happiness, surprise, sadness, anger, disgust,* and *fear*). We follow Shen et al. (2021) to regard utterance turns as speaker turns.

**MELD** (Poria et al., 2019): A multimodel ERC dataset with 7 emotion labels as the same as Daily-Dialog.

**EmoryNLP** (Zahiri and Choi, 2018): A TV show scripts dataset with 7 emotion labels (*neutral, sad, mad, scared, powerful, peaceful, joyful*).

**IEMOCAP** (Busso et al., 2008): A multimodel ERC dataset with 9 emotion labels (*neutral, happy, sad, angry, frustrated, excited, surprised, disappointed,* and *fear*). However, models in ERC field

are often evaluated on samples with first six emotions due to the too few samples of latter three emotions. 20 dialogues for validation set is following Shen et al. (2021).

**RECCON** (Poria et al., 2021): The first dataset for emotion cause recognition of conversation including RECCON-DD and RECCON-IE (a subset emulating an out-of-distribution generalization test). RECCON-DD includes 5380 labeled ECPs and 5 cause spans (*no-context*, *inter-personal*, *self-contagion*, *hybrid*, and *latent*).

**Synthetic dataset**: We create a synthetic dataset by following the benchmark of the causal discovery field (Agrawal et al., 2021; Squires et al., 2022). To minimize sample bias, we did not randomly draw causal graphs as samples. Inversely, the number of samples in the synthetic dataset and the number of utterances and labels per sample are restricted to be consistent with RECCON. We use Causal Additive Models (CAMs), Specifically SCM structure for our datasets. As shown in Algorithm 7, first, we assume that each $i.i.d.$ implicit causes $E \sim \|^{50}\mathcal{N}(1,1)$ if it is an emotion utterance in the original dataset, and $E \sim \|^{50}\mathcal{N}(-1,1)$ if it is not. Then, we update each utterance via speaker turns $S$: if there is an emotion-cause pair $(U_i, U_j) \in L$, then $U_i = \alpha_{j,i}U_j + E_i$ ($\alpha_{j,i} \sim Unifrom([0.7,1])$), and for those pairs without emotion-cause label, $\alpha_{j,i} \sim Unifrom([0,0.3])$. Finally, we randomly select a noise $\xi \sim Unifrom([-0.25, 0.25])$ for each utterance $U_i = U_i + \xi_i$.

## D  Implementation Details

In the word embedding, we adopt the affect-based pre-trained features[1] proposed by Shen et al. (2021) for all baselines and models.

Although there are different pre-trained models in these skeleton baselines, the SOTA work DAG-ERC and EGAT have investigated their performances in a consistent pre-trained model. Therefore, for a fair and direct comparison, we continue this benchmark using the pre-trained embedding published by DAG-ERC for three tasks.

In the hyper-parameters, we follow the setting of Shen et al. (2021) in the ERC task. Moreover, in the ECPE and ECSR, the learning rate is set to 3e-5, batch size is set to 32, and epoch is set to 60. Further in our approach, we set $L$ to 1, and implicit cause size is set to 192, hidden size of GNN is set

---

**Algorithm 7:** Creating Non-noise Synthetic dataset

**Input:** $\mathcal{D}, S, L$
**Output:** $SCM_{\mathcal{D}}$

1 **forall** $i \in 2, 3, \ldots, S$ **do**
2    **if** $Emotion(U_i)$ **then**
3       $E_i \sim \|^{50}\mathcal{N}(1,1)$
4    **else**
5       $E_i \sim \|^{50}\mathcal{N}(-1,1)$
6    $U_i \leftarrow E_i$

7 **forall** $i \in 1, 2, 3, \ldots, S$ **do**
8    **forall** $j \in 1, 2, \ldots, i$ **do**
9       **if** $(U_i, U_j) \in L$ **then**
10          $U_i = \alpha_{j,i}U_j + E_i(\alpha_{j,i} \sim$
         $Unifrom([0.7,1]))$
11       **else**
12          $U_i = \alpha_{j,i}U_j + E_i(\alpha_{j,i} \sim$
         $Unifrom([0,0.3]))$

13 $SCM_{\mathcal{D}} \leftarrow U_1, U_2, \ldots, U_S$ return $SCM_{\mathcal{D}}$

---

to 300, and dropout rate is 0.3.

Meanwhile, because there is only one training dataset for ECPE and ECSR, we evaluated our method ten times with different data splits by following Chen et al. (2023) and then performed paired sample $t$-test on the experimental results.

Finally, we adopted downstream task modules consistent with the SOTA baselines: Wei et al. (2020) in ECPE and ECSR, and Shen et al. (2021) for the ERC task.

For evaluation metrics, we follow Shen et al. (2021) towards ERC, Xia and Ding (2019) towards ECPE, and Poria et al. (2021) towards ECSR. Specifically, we adopt the macro F1 score in ECPE and ECSR tasks, micro F1 score for DailyDialog, and macro F1 score for the other three datasets in ERC task.

## E  Other Experiments in Affective Reasoning

In Table 7, our approach performs better than the corresponding baseline under all skeletons in four datasets. Hence, using a causal auto-encoder to find the implicit causes benefits this task. Besides, our approach improves significantly under skeletons **II**, **III**, and **IV**. From Figure 2, these three skeletons have more relevant nodes than others, so there are more redundant edges to be corrected

---

[1] https://drive.google.com/file/d/1R5K_2PlZ3p3RFQ1Ycgmo3TgxvYBzptQG/view?usp=sharing

| Skt | Model | DailyDialog | MELD | EmoryNLP | IEMOCAP |
|---|---|---|---|---|---|
| II | DialogXL | 54.93 | 62.41 | 34.73 | 65.94 |
| | Ours | **59.51** | **63.62** | **39.16** | **66.47** |
| III | EGAT† | 59.23 | 63.51 | 38.77 | 66.76 |
| | Ours | **59.68** | **63.71** | **39.62** | **68.18** |
| IV | RGAT | 54.31 | 60.91 | 34.42 | 65.22 |
| | Ours | **59.65** | **63.69** | **39.22** | **67.65** |
| V | DECN† | 59.08 | 63.78 | 39.44 | 67.41 |
| | Ours | **59.28** | **63.91** | **40.11** | **67.61** |
| VI | DAG-ERC | 59.33 | 63.65 | 39.02 | 68.03 |
| | Ours | **59.53** | **63.81** | **39.54** | **69.17** |

Table 7: Overall performance in ERC task. † denotes the results implemented in this paper. The better scores in the same skeleton are in bold, and the best of all skeletons is in red.

| Skt | DailyDialog | MELD | EmoryNLP | IEMOCAP |
|---|---|---|---|---|
| II | 51.48 ($\downarrow$8.03) | **58.41** ($\downarrow$**5.21**) | 34.97 ($\downarrow$4.19) | 59.71 ($\downarrow$6.76) |
| III | 54.37 ($\downarrow$5.31) | 58.19 ($\downarrow$5.52) | 36.55 ($\downarrow$3.07) | 63.42 ($\downarrow$4.76) |
| IV | **55.62** ($\downarrow$**4.03**) | 57.22 ($\downarrow$6.47) | **36.91** ($\downarrow$**2.31**) | 62.34 ($\downarrow$5.31) |
| V | 54.62 ($\downarrow$4.66) | 58.19 ($\downarrow$5.72) | 35.49 ($\downarrow$4.62) | 63.13 ($\downarrow$4.48) |
| VI | 53.27 ($\downarrow$6.26) | 58.39 ($\downarrow$5.42) | 34.98 ($\downarrow$4.56) | **65.18** ($\downarrow$**3.99**) |

Table 8: Overall performance of implicit causes $E$ in ERC task.

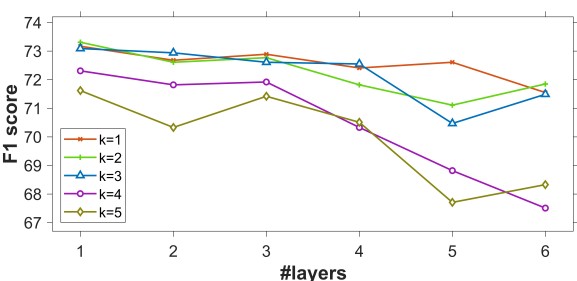

Figure 6: Further layers $L$ and related node number $K$ with **VI** skeleton model in ECPE task.

by our approach, which is demonstrated again in Appendix E. In contrast, **V** and **VI** achieve the best results in MELD, EmoryNLP, and IEMOCAP datasets, which indicates that **Hypothesis 5** is more probably a strong inductive bias that conversation enjoys.

Then, we investigate how the number of layers and the variants of causal skeletons would affect the performance of our approach. So we further conducted several contrasts with $k$ up to 5 and $L$ up to 6, as shown in Figure 6. One observation is that the best performance occurs at either $k = 1, 2$, or 3, which indicates that $k \geqslant 4$ offers no advantage and even leads to confounding. Moreover, $L = 1$ achieves the best performance under all $k$ values. In other words, one layer is sufficient to yield the most effective implicit causes.

## F  Visualization of Causal Graph

In the Figure 7 to 11, we showed the Visualization of the adjacency matrix $(I - A^T)^{-1}$. When the auxiliary loss $Loss_{KL}$ achieves the lower bound, $(I - A^T)^{-1}$ represents the relationship matrix between utterances and implicit causes.

In the ECPE task, we extracted 10 samples from test sets in different folds. To facilitate comparison and contrasting, we selected five 7-utterances cases and five 8-utterances cases. The IDs are as

following:

**7-utterances cases**: 110, 170, 224, 372, 500.

**8-utterances cases**: 62, 74, 104, 177, 584.

To obtain the non-negative value, we adopted the $T = sigmoid(\cdot) - 0.05$ to process the original tensors $(I - A^T)^{-1}$ outputted from the encoder. We follow a common practice: set the threshold as $0.05$ to delete some unimportant edges. And to highlight which implicit cause contributes the each utterance best, we adopted the $softmax(\cdot)$ to process columns afterward and labeled the block with value $> 0$.

It is excepted that: (i) when skeletons construct overage edges, our model is able to degrade the influences of some negligible utterances by deleting the corresponding edges from their implicit causes. (ii) when skeletons construct insufficient edges, our model can add some edges to obtain more information.

## G  Proof of emotion consistency of implicit causes and utterances

We would like to explain why implicit causes and utterances are consistent in emotion from both theory and euqation, in the condition where emotional utterance and cause utterance possess different emotion types.

We define the implicit causes as the unobservable emotional desire and the utterances as the observable emotional expression. This definition is proposed in Ong et al. (2019, 2015), which also argues that emotional expression is affected by desires and event outcomes. Moreover, for emotion utterances that are not influenced by explicit cause factors, the source of their emotions should originate from implicit causes. The desire and the expression generally belong to the same emotion because the outcomes often have little effect on emotional expression. Our paper can also deduce this conclusion from the SCM (Equation 1). Considering there is a linear map $f(\cdot)$ from representation space to emotion space. Then we can obtain the

following:

$$f((I - A)U) = f(E) \qquad (16)$$

$$(I - A)f(U) = f(E) \qquad (17)$$

$$f(U) = W^T f(E) \qquad (18)$$

Note that $W = (I - A)$ and $A_{i,i} = 0$. So in $W$, the value of the elements on the diagonal is constant at $1$ and is a constant maximum of each column. Naturally, $f(E)$ is an approximate estimate of $f(U)$ especially $U_t$ and $U_i$ in the ECP $(U_t, U_i)$ do not belong to the same emotion category, which is why we think implicit causes are reasonable when the F1 score of Table 6 is high.

Therefore, we test the F1 scores in ERC task by replacing $\widehat{H}$ with $E$ from a consensus that implicit causes should be aligned with utterances in the emotion types.

In Table 8, we reported the overall results of $E$ in ERC task. Note that we only examine the sample of ECP with different emotion types. Among five skeletons and four datasets, almost all results achieve $90\%$ scores of corresponding performances of $\widehat{H}$, which indicates that $E$ is practically aligned with $\widehat{H}$ in the affective dimension.

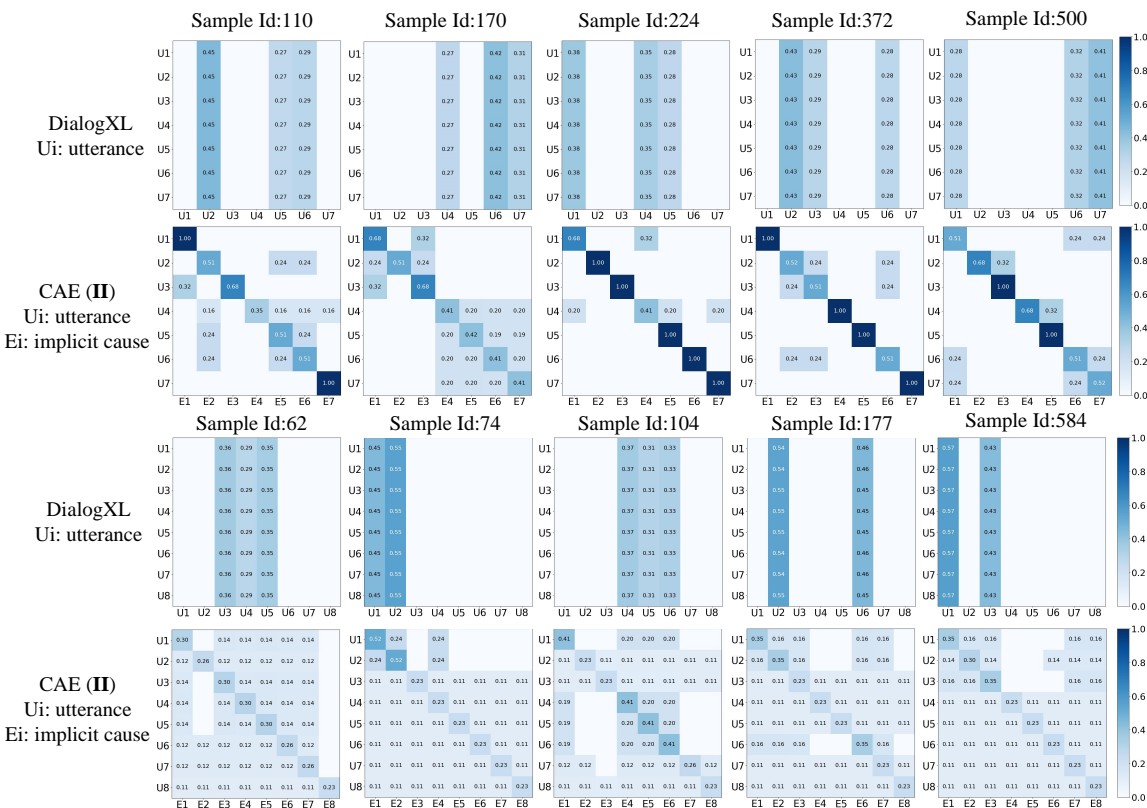

Figure 7: Causal Graph cases of DialogXL and Ours (CAE **II**).

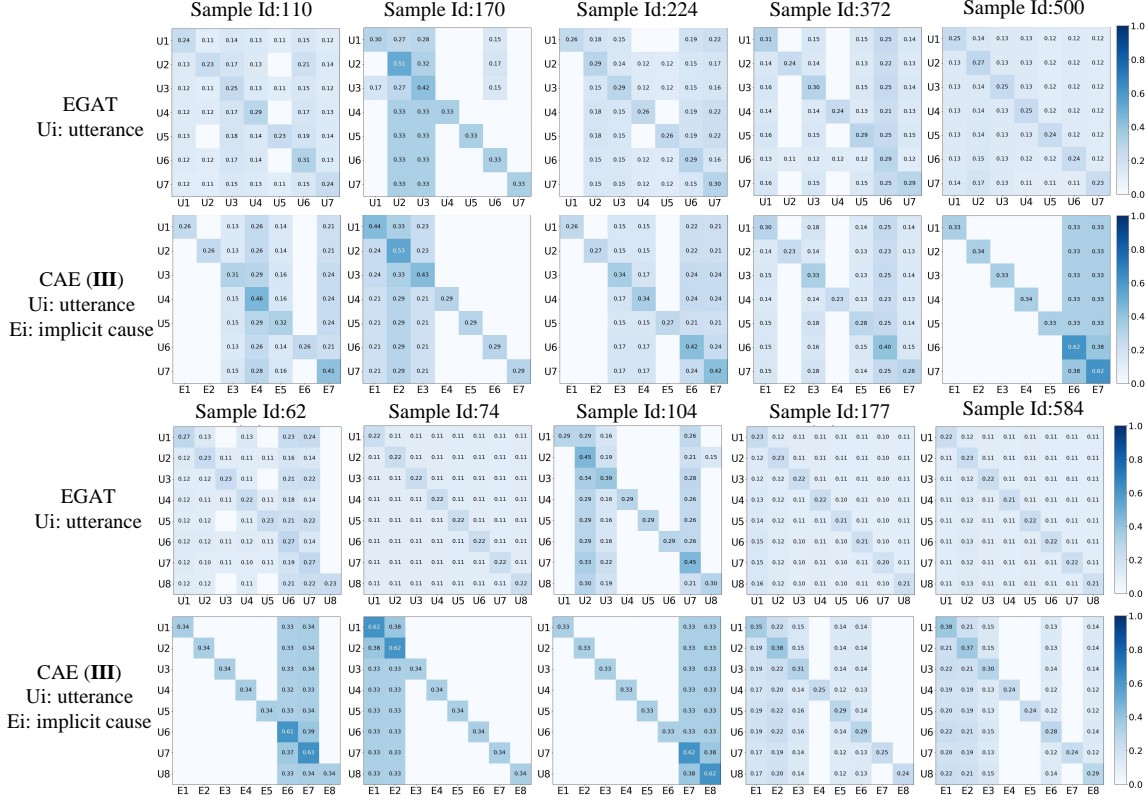

Figure 8: Causal Graph cases of EGAT and Ours (CAE **III**).

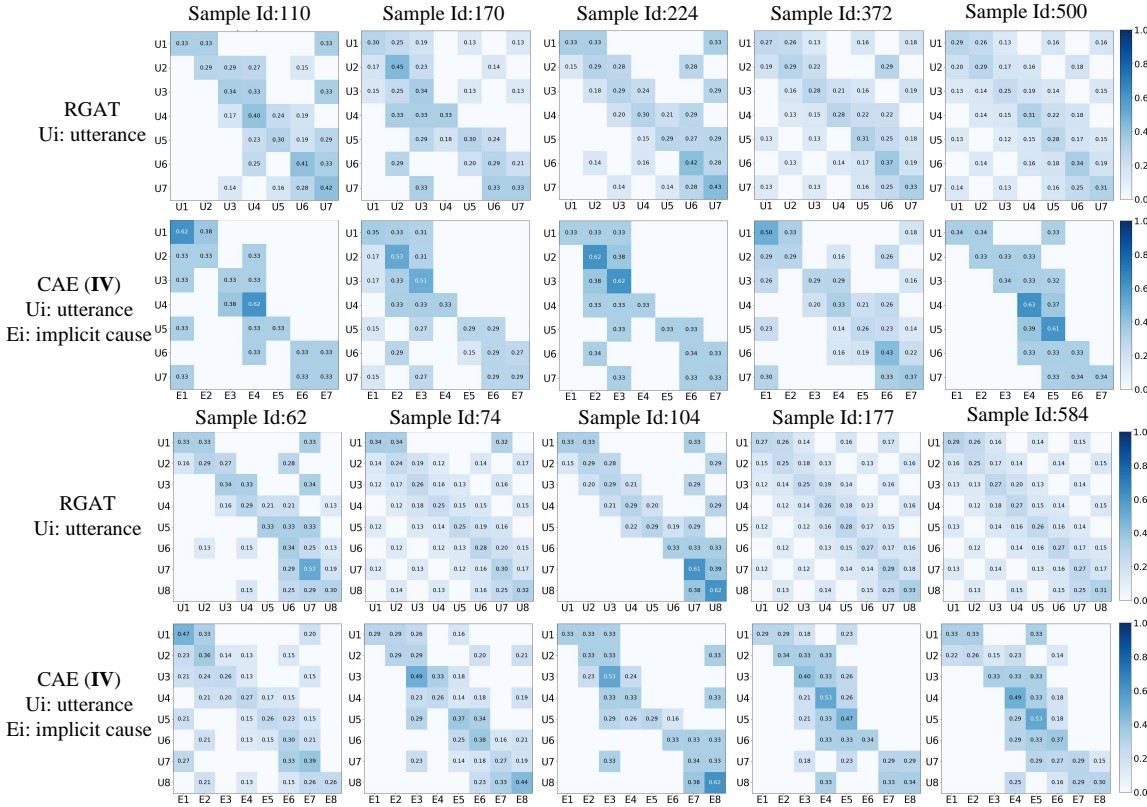

Figure 9: Causal Graph cases of RGAT and Ours (CAE **IV**).

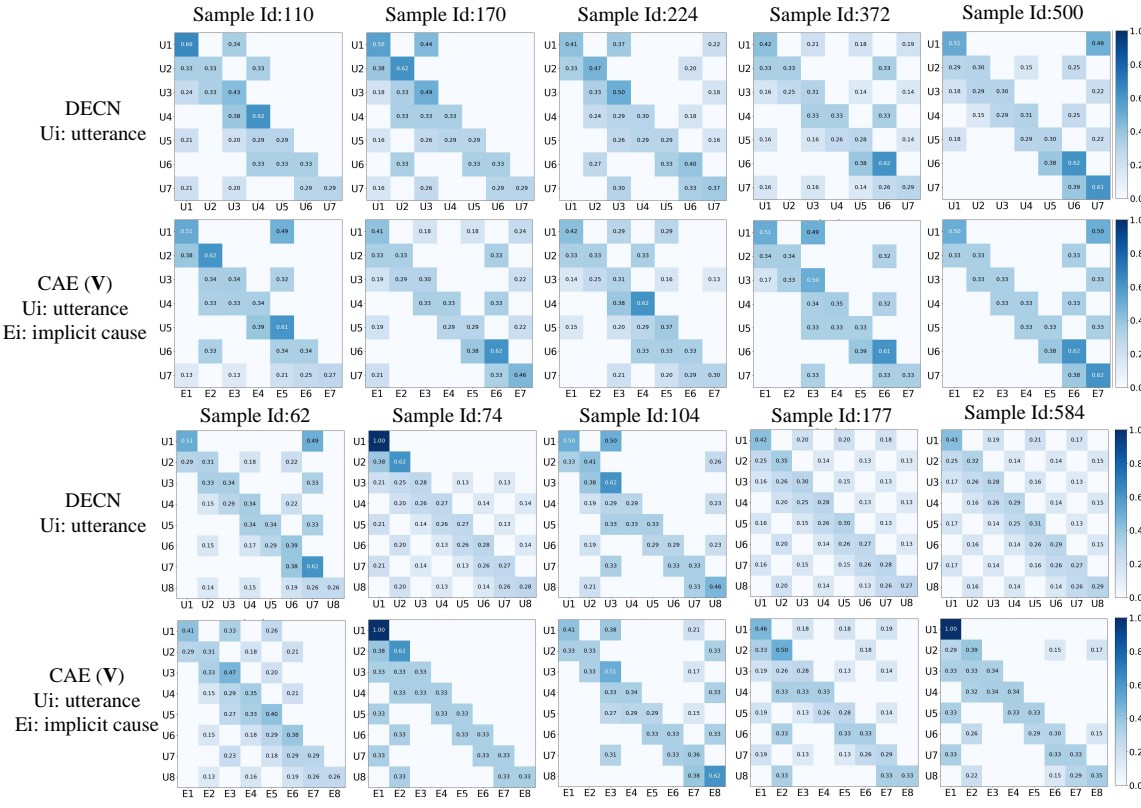

Figure 10: Causal Graph cases of DECN and Ours (CAE **V**).

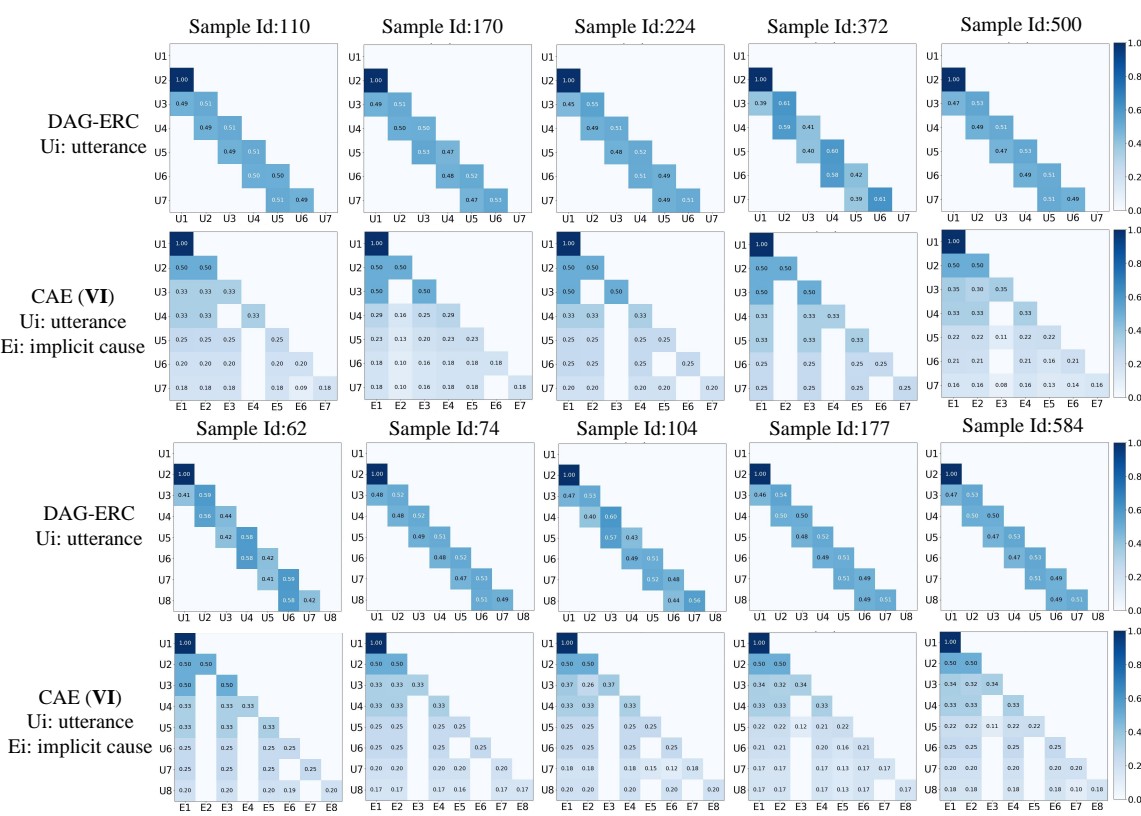

Figure 11: Causal Graph cases of DAG-ERC and Ours (CAE **VI**).