# OpenReview forum: "How to Enhance Causal Discrimination of Utterances: A Case on Affective Reasoning"
_EMNLP/2023/Conference — EMNLP 2023 Main_

### Official Review · Reviewer_28nr · 2023-08-04

**Paper Topic And Main Contributions:** 1. The authors demonstrate the effect…
**Soundness:** 4

**Excitement:**

5: Transformative: This paper is likely to change its subfield or computational linguistics broadly. It should be considered for a best paper award. This paper changes the current understanding of some phenomenon, shows a widely held practice to be erroneous in someway, enables a promising direction of research for a (broad or narrow) topic, or creates an exciting new technique.

**Reasons To Accept:**

1. This paper is well-organized, clearly describing the task and problem.

2. The proposed method is based on a high level of understanding of existing research and includes the definition of a valid hypothesis with technical descriptions.

3. It clearly describes the limitations and provides appropriate analysis through experiments.

**Reasons To Reject:**

1. The authors have partly addressed this in the limitations section, but considering confounding-bias and the backdoor problem is relatively weak compared to previous studies on causality.

2. Isn't the proposed method also not a fundamental solution to the challenge of affective reasoning? If there are labels and structures that can represent causality in a given conversation situation, I think it is natural to see the effect of investing more layers and resources.

**Reproducibility:**

3: Could reproduce the results with some difficulty. The settings of parameters are underspecified or subjectively determined; the training/evaluation data are not widely available.

**Reviewer Confidence:**

3: Pretty sure, but there's a chance I missed something. Although I have a good feel for this area in general, I did not carefully check the paper's details, e.g., the math, experimental design, or novelty.

**Typos Grammar Style And Presentation Improvements:**

Figure 3. The description of the colors and shading is missing and hard to see, which is confusing.

---

> ### Author Rebuttal · Authors · 2023-08-23
>
> Thank you for the time and effort you have invested in reviewing our paper. From your comments, it is evident that you have carefully scrutinized our work, and many of your concerns align closely with our latest research. We would like to address each of your points and clarifications in turn:
>
> Point 1: The limitations of confounding bias and backdoor paths.
>
> We have been attempting to integrate previous methods of causal research into dialogue and other deep-learning inference tasks. To overcome these limitations, in our new work, we have proposed interventions suitable for deep representations. Essentially, traditional interventions $do(X):= P(X = x_{1}) = 1$, but our proposed interventions for deep representations are defined as $do(X):= Pa(X) = \emptyset $, where $Pa(X) $ denotes the parent set of $X$. We hope you will soon see the details of this insight in our soonly papers.
>
> Point 2: Address the challenge more fundamentally.
>
> The most significant obstacle to further research is the lack of a high-quality dataset with complete causal relationship labels. Therefore, we have constructed a simulated dialogue dataset and plan to make it open soon. The new dataset, generated via the GPT-4 API, controls the preamble utterances that can be known. Simultaneously, to resolve this issue, we are considering many new aspects, such as determining the causal consistency of causal structures and representations. This guiding thought leads us to consider the causal abstraction in dialogue causal relationships, which aligns with the novel intervention insights mentioned in point 1. In subsequent manuscript versions, we will add these insights in the limitations section.
>
> Point 3: Suggestions on writing.
>
> We appreciate your diligent review. In future versions, we plan to provide more comprehensive explanations for the details in figures and tables.

---

### Official Review · Reviewer_2DCP · 2023-08-08

**Soundness:** 4

**Excitement:**

3: Ambivalent: It has merits (e.g., it reports state-of-the-art results, the idea is nice), but there are key weaknesses (e.g., it describes incremental work), and it can significantly benefit from another round of revision. However, I won't object to accepting it if my co-reviewers champion it.

**Paper Topic And Main Contributions:**

This paper proposes an approach to "affective reasoning in conversation" formulated as a "causal discrimination" task, which essentially aims to classify whether a pair of conversation turns has a causal relationship (and if so, in what directionality).

The underlying premise of this work (which is assumed but not formally or empirically demonstrated) is that this particular task formulation of emotional causal relation classification in a pair of conversation turns is important.

The key thesis of this paper is that handling of i.i.d. noise is important in reasoning about such structural causal relationships, and the paper thus proposes an autoencoder architecture that can incorporate such i.i.d. noise, which presumably corresponds to the "implicit causes".

**Reasons To Accept:**

- the paper addresses causal reasoning, an important and open research challenge in today's AI literature and proposes an approach for structural causal reasoning, demonstrating stronger performance over baselines

**Reasons To Reject:**

- I wish the paper provided a better description of the task itself. Based just on Figure 1 example, I couldn't help but wonder if the correct answer has a trivial pattern --- that there's a causal relationship between the most recent utterance of between two speakers (100% cases?), and also that there's a causal relationship between two most recent utterances of each speaker (the vast majority of the cases?), which makes me wonder whether simple majority baselines of some sorts could've worked surprisingly well. Detecting when the causality misses between two adjacent utterances (which can happen, though less frequently) seems relatively easy due to the abrupt change of the topic. Fundamentally, I came away with the suspicion --- whether the classification formulation experimented in this paper justifies the relatively complex method proposed in this work, and I wish there was a more convincing benchmark explored in this paper.

- Concurrently, I'm not convinced if the relatively simplistic classification of causal relationship explored in this paper helps advancing the field in terms of affective understanding in conversation, especially in the context of the currently powerful LLMs that can carry about an amazing level of conversation. While causal reasoning is generally an interesting and open research question, I wonder if the particular task formulation explored in this paper is rather too narrow to be relevant in the current AI SOTA capabilities.

- I wish the authors provided more intuition about why "implicit causes" correspond to "i.i.d. noise". While I understand to some degree that implicit causes may seem like "noise", various statements around this don't seem well justified or even precisely stated.

- The lack of clarity of writing is among the weakest points of this paper. At least for me, of the couple dozen papers across NLP and ML venues that I have reviewed in recent years, this one by far was the most opaque to unpack. As one of the many such examples, I found the following sentence hard to parse and hard to fully understand [line 073 - line 079]: "In order to discriminate different causal relationships between two similar embeddings, we construct the dialogue process as a Structural Causal Model (SCM) stemming from many endeavors supporting that i.i.d. noise of SCM could facilitate the discrimination of causal relationships when finning two variables." In fact, I found almost half of the remaining sentences of the introduction section somewhat troublesome, sometimes because the sentences make claims that are not yet defined or justified.

- In the same vein, the technical writing seems a bit sloppy at times. For example in Table 3, DD and IE were never defined, and the caption talks about ECPE-MLL, while it doesn't even appear in the rows of the methods listed in that Table. Moreover, the boldfaced numbers seem inconsistent. For example, GPT-4's IE score is higher than that of ECPE-2D, but not boldfaced.

**Reproducibility:**

3: Could reproduce the results with some difficulty. The settings of parameters are underspecified or subjectively determined; the training/evaluation data are not widely available.

**Reviewer Confidence:**

3: Pretty sure, but there's a chance I missed something. Although I have a good feel for this area in general, I did not carefully check the paper's details, e.g., the math, experimental design, or novelty.

---

> ### Author Rebuttal · Authors · 2023-08-23
>
> Thank you for your detailed review. However, after going through your comments, we couldn't help but wonder if you may have missed or skipped many sections of the paper (especially section 3.2 and Appendix B). We understand this, considering that our paper, with its appendix, exceeds the average length for a conference paper, so it's possible that you had to overlook some parts that you deemed less critical or content that falls into the appendix, given the review deadline. Regardless, we will respond to each of your points and indicate the corresponding sections where we have provided sufficient clarification:
>
> Point 1: More convincing benchmarks with simple patterns.
>
> We have explicitly introduced these 'simple patterns' in our paper. Specifically, in Section 3.2 and Appendix B, we formulate Hypotheses 1-6 and Skeletons 1-6 to describe these simple patterns (called prior knowledge in our papers). Moreover, in the experiments, we indeed treat the SOTA works on these simple patterns as benchmarks. Further, there are some inaccuracies and incomplete observations in the simple patterns you mentioned. For instance, in $U_{1}$: ''Let's name a red object'', $U_{2}$: ''Red Apple'', $U_{3}$: ''Red Sun'', a 100% causal connection between $U_{2}$ and $U_{3}$, as you mentioned, is not present. The relationship between ''Red Apple'' and ''Red Sun" is of similarity rather than causality, the common reason being $U_{1}$. That is, the causal relationship is $U_{3} \leftarrow U_{1} \rightarrow U_{2}$. For other simple patterns you may not have considered, we invite you to refer in detail to Appendix B.
>
> Point 2: You believe that the strong capabilities of LLM undermine our contribution.
>
> Our Tables 1, 3, and 5 emphasize that although LLMs have strong abilities, they lack causal inference capabilities. Simultaneously, many recent studies have shown that LLMs lack causal discriminability [1][2]. The weak causal inference capacity results in those without extensive complex prompt instructions, performing worse than supervised specialized models. Therefore, for these tasks, we believe that the contribution deriving from improvements to supervised specialized models far outweighs mere research on LLM prompts and fine-tuning. Moreover, the need to construct unique skeletons for each specific domain is the reason why our paper only takes the dialog emotion inference task as an example. However, our contribution lies in: for most text inference tasks, it can introduce i.i.d. noise, follow our skeleton construction process, and then apply the model we propose to gain causal insights.
>
> Point 3: You are unclear as to why implicit causes can correspond to independent noise.
>
> We have explained this issue in the papers, and you could check lines 245-255 in Section 3.1 and lines 324-333 in Section 3.3. To explain simply: The insights in this part involve cognitive science[3] and conversation theory[4]. For each utterance, while it superficially appears to be directly influenced by the preceding utterances, the actual causing factor is mental states within the speaker, such as desires, goals, and beliefs. These mental states are also referred to as common ground in the field of cognitive science. Therefore, the preceding utterances, which we refer to as explicit causes, are like the parents  set of X in the SCM; they are endogenous and observable. Meanwhile, the mental states, which we term as implicit causes, are like the noise terms in the SCM; they are exogenous and unobservable. Hence, they can naturally correspond to each other.
>
> Point 4: Unclear writing.
>
> Thank you for your feedback. Although the other two reviewers found our manuscript to be clear and detailed, we value your perspective and will replace some of the longer sentences with more short ones in the final draft. Indeed, our paper covers a range of topics, which may make reading difficult for readers with less background in these areas.
>
> Point 5: Formatting errors.
>
> We appreciate your observations. Multiple updates to the paper have unfortunately led to some inconsistencies and errors. We will thoroughly correct these in upcoming versions. However, DD and IE are mentioned in Appendix C (line 967).
>
>
> [1]. Jin, Zhijing, et al. "Can Large Language Models Infer Causation from Correlation?." arXiv preprint arXiv:2306.05836 (2023).
> [2]. Long, Stephanie, et al. "Can large language models build causal graphs?." arXiv preprint arXiv:2303.05279 (2023).
> [3]. McKinley, Geoffrey L., Sarah Brown-Schmidt, and Aaron S. Benjamin. "Memory for conversation and the development of common ground." Memory & cognition 45 (2017): 1281-1294.
> [4]. Chen, Hang, et al. "Learning a Structural Causal Model for Intuition Reasoning in Conversation." arXiv preprint arXiv:2305.17727 (2023).

---

### Official Review · Reviewer_5DgJ · 2023-08-11

**Soundness:** 4

**Excitement:**

3: Ambivalent: It has merits (e.g., it reports state-of-the-art results, the idea is nice), but there are key weaknesses (e.g., it describes incremental work), and it can significantly benefit from another round of revision. However, I won't object to accepting it if my co-reviewers champion it.

**Paper Topic And Main Contributions:**

This study concentrates on the challenge of causal discrimination, specifically on the task of affective reasoning in *conversations*. The paper reveals that almost all existing models excel at capturing semantic relatedness within utterance embeddings but fall short in determining the specific causal relationship underlying the association between two utterances on the task of causal discrimination. To overcome this limitation, the authors constructed a SCM with i.i.d. noise terms, and proposed the *cogn* framework to address the unstructured nature of conversation data. Experimental results show that the proposed approaches significantly outperform existing methods on two affective reasoning tasks including Emotion-Cause Pair Extraction (ECPE) and Emotion-Cause Span Recognition (ECSR) and one emotion recognition task (ERC), demonstrating its effectiveness in affective reasoning.

The main contributions are as follows: 1) The authors incorporated i.i.d. noise terms, and formulated the dialogue process as a structural causal model (SCM); 2) The authors devised the *cogn* skeleton to address the problems of variable-length and unstructured dialogue samples; 3) The authors adopted an autoencoder architecture to overcome the unobservability of implicit causes and make it learnable; and 4) The authors constructed a synthetic dataset with implicit causes and conducted extensive evaluations of the proposed method.

**Reasons To Accept:**

1. The study reveals that when it comes to more specific causal relationships within semantically similar sentences (such as reasoning tasks), both unsupervised and supervised methods may not exhibit the same level of “intelligence” and output some “pseudo-correlation”.
2. The discussion of the methods proposed in this work is thorough and detailed.
3. The authors propose new methods to address the problem of causal discrimination in affective reasoning. Their approaches significantly outperform other state-of-the-art methods, including GPT-3.5 and GPT-4.

**Reasons To Reject:**

There are too many abbreviations throughout the paper (e.g., ECP, ERC, ECPE, ECSR), which can be a bit confusing sometimes. Otherwise, I see no reasons to reject this paper.

**Reproducibility:**

4: Could mostly reproduce the results, but there may be some variation because of sample variance or minor variations in their interpretation of the protocol or method.

**Reviewer Confidence:**

3: Pretty sure, but there's a chance I missed something. Although I have a good feel for this area in general, I did not carefully check the paper's details, e.g., the math, experimental design, or novelty.

**Typos Grammar Style And Presentation Improvements:**

There are too many abbreviations throughout the paper (e.g., ECP, ERC, ECPE, ECSR), which can be a bit confusing sometimes.

---

> ### Author Rebuttal · Authors · 2023-08-23
>
> We sincerely appreciate Reviewer 5Dgj's attention to the details in our paper. We fully agree with your perspective and apologize for our oversight regarding abbreviations. Particularly for ECPE, ECSR, and ECP, it can be challenging for many readers to distinguish between them during their initial reading.
>
> To address your concerns, we will make the following changes in the final version: whenever an abbreviation appears for the first time in each section or page, we will provide its full form. In this way, we hope readers can easily find the full forms whenever they feel uncertain about the abbreviations. Alternatively, we may use more distinctive abbreviations that better represent each term, such as 'ECP' = 'EC pairs', 'ECPE' = 'EC pairs extraction', and 'ECSR' = 'EC span recognition'.
>
> Lastly, thank you for investing your time and effort in reviewing our paper.

---

### Meta-Review · Area_Chair_S3yp · 2023-09-19

**Recommendation:** 4

**Metareview:**

The paper presents a narrow and focused contribution to affective reasoning in conversations through the lens of causal discrimination, predicting the presence of causal relationship and direction in pairs of utterances. The paper then presents structural causal model with the  i.i.d. noise inclusion into conversation. To incorporate such i.i.d. noise, the paper presents an Autoencoder framework.  The study of causal relationship (in particular for LLMs or not-so-large LMs) is very important but much less explored compared to the rest of the field. As such, this work provides contributions, although very narrow in its task scope, which are interesting and insightful. The presentation of the paper could be polished further (following the comments of 5DgJ, 2DCP).  The abstract and presentation of results could be decompressed further.

---

### Decision · Program_Chairs · 2023-10-07

**Decision:**

Accept-Main

**Comment:**

The paper presents a narrow and focused contribution to affective reasoning in conversations through the lens of causal discrimination, predicting the presence of causal relationship and direction in pairs of utterances. The paper then presents structural causal model with the  i.i.d. noise inclusion into conversation. To incorporate such i.i.d. noise, the paper presents an Autoencoder framework.  The study of causal relationship (in particular for LLMs or not-so-large LMs) is very important but much less explored compared to the rest of the field. As such, this work provides contributions, although very narrow in its task scope, which are interesting and insightful. The presentation of the paper could be polished further (following the comments of 5DgJ, 2DCP).  The abstract and presentation of results could be decompressed further.